# Medaka (*Oryzias latipes*) initiate courtship and spawning late at night: Insights from field observations

**Yuki Kondo**[1,2]*, **Kotori Okamoto**[2], **Yuto Kitamukai**[1], **Yasunori Koya**[3], **Satoshi Awata**[1,2]

**1** Laboratory of Animal Sociology, Department of Biology, Graduate School of Science, Osaka Metropolitan University, Osaka, Japan, **2** Laboratory of Animal Sociology, Department of Biology and Geosciences, Graduate School of Science, Osaka City University, Osaka, Japan, **3** Department of Biology, Faculty of Education, Gifu University, Gifu, Japan

* youkikondou@gmail.com

**Data Availability Statement:** All relevant data are within the manuscript and its Supporting Information files.

## Abstract

Laboratory experiments were conducted using model organisms to elucidate biological phenomena. However, the natural habitats of organisms are inherently more complex than those found in the laboratory. To complement the laboratory experiments, we conducted field observations of the small freshwater fish medaka (*Oryzias latipes*), widely used as a model organism, to elucidate its ecology and behavior in natural environments. Our results showed that medaka initiated courtship and spawning late at night, much earlier than previously thought. Nocturnal video observations examining spawning time during the breeding season in Gifu, Japan (sunset: 19:00; sunrise: 5:00) revealed the presence of post-spawning medaka females around midnight. Behavioral analysis showed that the medaka was inactive until 23:00, with activity increasing from 0:00 and peaking from 1:00 to 3:00. Furthermore, a significant increase in male courtship was observed between 0:00 and 4:00. These findings provide the first empirical evidence that medaka mating begins significantly earlier than previously reported in the laboratory, as within an hour before or after light onset in the morning. This study highlights the importance of field observations in revealing critical aspects of organismal biology that may be overlooked in laboratory settings.

## Introduction

Model organisms are representative species used for research in various biological fields. Numerous studies using model organisms have been conducted in laboratories, contributing to our understanding of complex biological phenomena in simple environments by controlling complicated conditions that are difficult to regulate. However, natural environments in which organisms live are not as simple as those in the laboratory [1]. Surprisingly, relatively little is known about the ecology, behavior, reproduction, and life histories of some model organisms in their natural environments, and the extent to which laboratory observations are consistent with their ecology and behavior in nature remains unclear. To comprehensively understand an organism, it is essential to elucidate its ecology and behavior in natural habitats

**Funding:** This study was funded by the Japan Society for the Promotion of Science (JSPS: https://www.jsps.go.jp/) KAKENHI (22K20666 to Y. Kondo and 23H03868 to S. A.), Sasakawa Scientific Research Grant from the Japan Science Society (2023-5013 and 2024-5010 to Y. Kondo: https://www.jss.or.jp/ikusei/sasakawa/), and the Tokyo Zoological Park Society Wildlife Conservation Fund (to Y. Kondo: https://www.tokyo-zoo.net/fund/index.html). The funders had no role in the study design, data collection and analysis, decision to publish, or preparation of the manuscript.

**Competing interests:** The authors have declared that no competing interests exist.

[2]. Field research provides insights into the ecology and behavior of the studied species. It contributes to understanding the adaptive significance and mechanisms of their genomic, morphological, and physiological traits. Understanding ecology and behavior in the wild also enables the maintenance, breeding, and experimentation of research species under conditions closer to their natural environment without inducing stress [3], consequently leading to improved animal welfare. Furthermore, this is particularly important to ensure the validity of the research outcomes, as the poorly understood welfare of the study species can affect the reliability of the findings.

The freshwater fish medaka (*Oryzias latipes*) is widely used as a model organism in various research fields, including physiology, genetics, developmental biology, behavioral science, and medicine. This is due to its unique combination of characteristics, such as small body size (3–4 cm standard length), ease of laboratory breeding, sexual dimorphism, short generation time, large and transparent eggs, and a relatively compact genome [4, 5]. However, despite using medaka as a model organism for over a century, its ecology and behavior in the wild, particularly its reproductive ecology, remain unclear [6]. Medaka (*O. latipes*) are distributed throughout Japan, inhabiting still or slow-moving water bodies such as rice paddies, ponds, and agricultural waterways [4]. The preferred habitat for medaka is shallow waters near the shore with abundant vegetation, where they feed on microorganisms, including algae and zooplankton [7]. Their lifespan in the wild is approximately one year [8]. The breeding season of medaka spans from early summer to autumn [9–12]. To our knowledge, mating behavior in the wild has only been reported by Kobayashi et al. (2012), in which, based on visual observations, mating behavior was observed primarily from early morning after sunrise to mid-morning [13].

In the laboratory, medaka mating is observed every morning, with males mating multiple times and females spawning once daily [14, 15]. Spawning typically occurs in pairs, and eggs are fertilized externally. After spawning, the female carries fertilized eggs on her abdomen and brushes against the aquatic plants to attach the eggs after a few hours. Many laboratory experiments on mating in medaka have been conducted because of the ease of observing their mating behavior. Examples include female mate choice [16–19], male mate-guarding behavior [20–22], alternative male reproductive tactics [23–25], and male sperm allocation [26, 27]. All studies were conducted under bright lighting conditions in the morning.

Interestingly, in the laboratory, females complete ovulation at night and are ready to mate in the morning [28]. Medaka activity levels increased significantly 4–5 h before the lights were turned on, which could be associated with behaviors such as searching for mating partners [29]. Furthermore, studies have reported that medaka can spawn in the dark [30] and mate without relying on vision [31]. Indeed, in our previous aquarium observations, females carrying eggs after spawning were frequently observed 3 h before the lights were turned on (Y. Kondo, personal observation). Accumulating evidence suggests that the onset of spawning in female medaka occurs before sunrise. If mating starts before light, the medaka breeding behavior observed under bright conditions in the laboratory may only cover a part of the breeding period (i.e., the latter part).

Therefore, this study aimed to elucidate the reproductive behavior and ecology of medaka during the breeding season in their natural habitat in Gifu, Japan. Based on the evidence of medaka reproduction, we used video cameras in streams during the night to observe behavioral changes. We determined the spawning onset time using egg-carrying females as indicators. Furthermore, we investigated changes in activity levels, courtship behaviors, and other mating behaviors throughout the night.

## Methods

### Study fish, study site, study period, and video recording

Medaka (*O. latipes*) is distributed in Japan, especially along the Sea of Japan coast west of Kyoto Prefecture and the Pacific coast south of Iwate Prefecture. It inhabits rivers, marshlands, and irrigation channels across lowland regions [4]. Based on the phylogenetic analyses of the cytochrome *b* gene, *O. latipes* was classified into 11 distinct clades [32]. Among these, the medaka targeted in this study were wild Gifu populations, which belong to subclades B-I, a group inhabiting the central region of Japan.

A field study was conducted in the small stream in July and August 2023 in Gifu, Japan (35˚47'16.22"N, 136˚76'46.64"E). Behavioral observations were performed with minimal water flow adjacent to the main stream where medaka were abundant. The average, minimum, and maximum temperatures during the study period were obtained from the Japan Meteorological Agency website (https://www.data.jma.go.jp/obd/stats/etrn/view/nml_sfc_ym.php?prec_no=52&block_no=47632&year=&month=&day=&view=), and the sunrise and sunset times were obtained from the National Astronomical Observatory of Japan website (https://eco.mtk.nao.ac.jp/koyomi/dni/2023/dni22.html) (S1 Table).

We conducted two field observation series using video recorders that recorded the spawning time and behavior of medaka. First, to confirm the time of spawning initiation, we conducted video recordings over four days from July 25 to July 28, 2023. A 20 cm × 20 cm quadrat was set up in the stream, and a video camera was positioned to capture the quadrat within its field of view. Four hours of video recording were conducted from 23:00 to 3:00 the following morning. Second, video recordings were conducted for ten days, from August 1 to August 10, 2023, to examine the behavior. Four 40 cm × 30 cm quadrats were set up in the stream. The recording time was approximately 13 hours daily, from 21:00 to 10:00 the following morning.

GoPro Hero 8 (GoPro Inc., CA, USA) with a 1 TB SD card (SanDisk, CA, USA) was used for recording. The video camera was connected to a mobile battery (Anker, Hunan, China) to enable long-duration recording, allowing continuous recording of over 24 hours. Field observations of medaka reproductive behavior have reported that red light is unlikely to affect reproductive activities [13]. Based on this finding, red light was used as the light source during the nighttime observations in this study. A red light (Explux Commercial Lighting, Tokyo, Japan) connected to a portable power source (Anker, Hunan, China) facilitated night recording. The mobile battery was stored in a cooler box with ice packs to prevent overheating, and a portable power source was placed in a bucket to avoid submersion.

### Behavior analysis

The video data were analyzed using the annotation software ELAN 6.1. First, we examined the data between 23:00 and 3:00 to determine the timing of spawning initiation. The video was divided into 10-minute segments, yielding 72 samples over four days. The videos were played in the laboratory, and the medaka entering the quadrat (20 cm × 20 cm) were analyzed. Female medaka spawned only once daily [15]. After spawning, females temporarily carry the spawned eggs hanging from their abdomens while swimming. All eggs were attached to aquatic plants or similar substrates within a few hours. This cycle was repeated on the following day [33, 34]. Therefore, the presence of a female swimming with eggs attached to her abdomen indicated that she had already completed spawning. Thus, checking whether eggs were attached to the female abdomen by the end of each 10-minute segment determined the occurrence or absence of mating before that time point. The time at which a female with eggs attached to her abdomen first appeared on the observation day was defined as spawning initiation time.

Second, we used four video cameras over ten days to examine behavior. We recorded videos from 21:00 to 10:00 the following day, but after sunrise (from approximately 5:00), it became challenging to observe the underwater world because of reflections on the water's surface; therefore, we were unable to analyze the behavior of the medaka from 5:00 to 10:00. For this reason, to analyze their behavior, we extracted 10-minute sample video segments from approximately 00 to 10 min each hour, spanning the period from 21:00 to 05:00 the following day (eight time points × four video cameras × nine recordings = 288 samples). However, 49 samples were excluded from the analyses because of heavy rain, resulting in a final count of 239.

The frequency of medaka entering the quadrat (40 cm × 30 cm) was counted as the number of appearances. The spawning behavior of medaka follows a sequence of events: the male chases the female (following), the male swims rapidly around the female (quick circle), the male wraps his dorsal and anal fins around the female (wrapping), the female releases eggs, the male releases sperm (egg and sperm release), and the male leaves the female (leaving). After spawning, the female carries eggs attached to her abdomen for several hours and rubs her abdomen against the aquatic plants to attach the eggs to the vegetation [33, 34]. Therefore, behaviors were categorized and quantified based on the following four classifications [15, 26, 34]: individual swimming (behaviors other than those listed below, used as indicators of medaka activity), staying (a state of not moving more than 5 cm in 5 s, presumably a resting situation), following (a courtship indicator in which a male chases a female), and a quick circle (a courtship indicator in which a male rotates in front of a female).

## Statistical analysis

All analyses and graphical illustrations were performed using R 4.3.2 (R Core Team, 2023). To investigate how the number of behaviors obtained from the video analysis changed from 21:00 to 5:00 the following day, generalized additive mixed models (GAMMs) were implemented using the *GAM* package. The significance of the models was evaluated using likelihood ratio tests with a significance level of $p < 0.05$.

First, we assessed whether the frequency of medaka appearances changed over time. A GAMM assuming a negative binomial distribution was created with the appearance frequency of the medaka set as the response variable, each period set as the fixed effect, and the video ID set as the random effect. Second, we examined how the frequency of the four medaka behaviors (individual swimming, staying, following, and quick circles) changed over time. The response variable was the number of medaka behaviors, and the explanatory variable was each period. Assuming a negative binomial distribution, they were created separately for each behavior, with the video ID set as a random effect.

## Ethical declarations

All methods used in this study were performed according to the relevant guidelines and regulations. The Animal Care and Use Committee of Osaka Metropolitan University (No. S0092) granted permission for the experiments. This study did not involve capturing or subjecting the wild medaka to lethal procedures. During the field study, efforts were made to minimize stress on the medaka.

## Results

### Initiation time of spawning

To determine the initiation time of spawning, video recordings were conducted over four days, from July 25 to 28, 2023. During the observations, multiple individual medakas were

visible in all videos. Egg-hanging females were observed at 23:54 on July 25, 0:47 on July 27, and 0:01 on July 28 (Table 1). Post-spawning females were consistently observed in all videos at 0:01 on July 28. On other recording days, post-spawning females were visible in nearly all videos from 1:40 to 1:50 am onwards (Table 1; S1 Movie).

## Nocturnal activity and courtship behavior

To examine the nocturnal activity and courtship behavior of the medaka, video analyses were conducted from 21:00 to 5:00 for ten days, from August 1 to 10, 2023. In all the videos analyzed, medaka appeared at least once in each video, with a total of 2,474 times (mean ± SE per one video = 10.36 ± 0.46, n = 239, S2 Table). The fewest appearances were recorded at 22:00 to 23:00, at 247 (mean frequency per video ± SE = 8.23 ± 1.20, n = 30), and the most at 1:00 to 2:00, at 377 (mean frequency per video ± SE = 12.57 ± 1.04, n = 30). The GAMM analysis revealed that the number of medaka appearances was similar from 21:00 to 23:00, gradually increased from 23:00, peaked from 0:00 to 2:00, and decreased after 2:00 (GAMM, deviance = -17.66, df = -5.60, p = 0.0054, n = 239; Fig 1).

Changes in individual swimming and staying times, which are indicators of activity, were examined between 21:00 and 5:00. Individual swimming was observed a total of 2,512 times (mean ± SE per one video = 10.51 ± 0.51, n = 239). The lowest frequency of swimming was recorded from 22:00 to 23:00, at 263 times in total (mean frequency per

**Table 1. Spawning time of medaka (*Oryzias latipes*) during the breeding season in Gifu, Japan.**

| Time | July 25th to 26th | July 26th to 27th | July 27th to 28th |
|---|---|---|---|
| 23:00–23:10 | 0 | 0 | 0 |
| 23:10–23:20 | 0 | 0 | 0 |
| 23:20–23:30 | 0 | 0 | 0 |
| 23:30–23:40 | 0 | 0 | 0 |
| 23:40–23:50 | 0 | 0 | 0 |
| 23:50–00:00 | 1(23:54) | 0 | 0 |
| 00:00–00:10 | 0 | 0 | 1(00:01) |
| 00:10–00:20 | 0 | 0 | 1 |
| 00:20–00:30 | 0 | 0 | 1 |
| 00:30–00:40 | 0 | 0 | 1 |
| 00:40–00:50 | 0 | 1(00:47) | 1 |
| 00:50–01:00 | 1 | 0 | 1 |
| 01:00–01:20 | 0 | 0 | 1 |
| 01:20–01:30 | 1 | 0 | 1 |
| 01:30–01:40 | 1 | 0 | 1 |
| 01:40–01:50 | 1 | 1 | 1 |
| 01:50–02:00 | 1 | 0 | 1 |
| 02:00–02:10 | 1 | 1 | 1 |
| 02:10–02:20 | 1 | 1 | 1 |
| 02:20–02:30 | 1 | 1 | 1 |
| 02:30–02:40 | 0 | 0 | 1 |
| 02:40–02:50 | 1 | 1 | 1 |
| 02:50–03:00 | 1 | 1 | 1 |

Videos where medaka females hanging eggs on their abdomens were observed are denoted with '1,' while those lacking such observations are marked with '0.' For videos capturing the first instance of a female with eggs hanging from her abdomen on that day, the observation time is shown in parentheses.

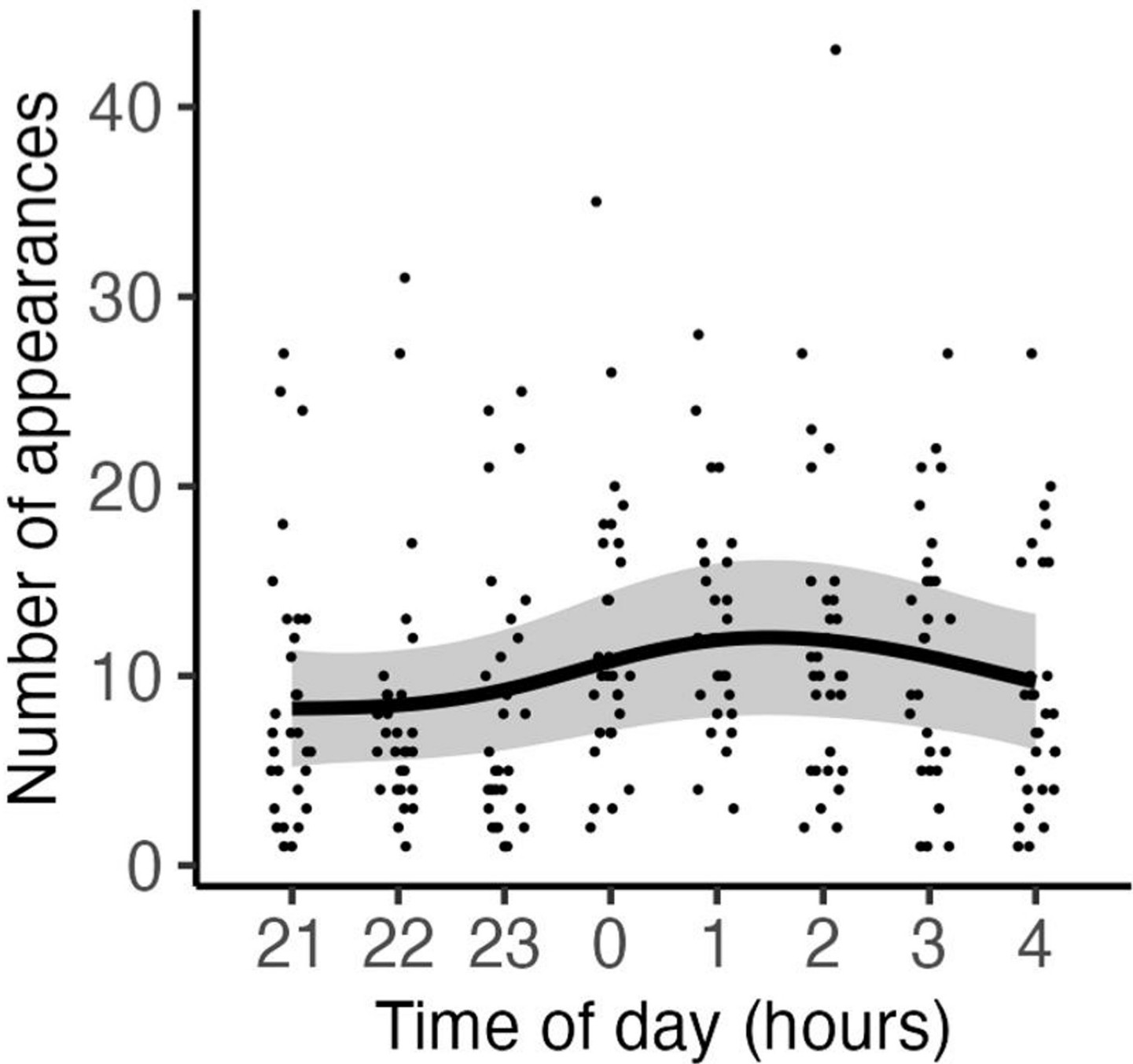

**Fig 1. Number of medaka (*Oryzias latipes*) appearances from 21:00 to 5:00 during the breeding season in Gifu, Japan.** Each plot signifies the observed values from the analyzed videos. The regression curve was based on the generalized additive mixed model (GAMM), and the gray shading indicates the 95% confidence intervals. 21:00–4:00: n = 30 for each, 4:00–5:00: n = 29 video samples.

video ± SE = 8.80 ± 1.56, n = 30), and the highest from 2:00 to 3:00, at 376 times (mean frequency per video ± SE = 12.53 ± 1.64, n = 30). Individual swimming gradually increased from 21:00 and remained high after 1:00 (GAMM, deviance = -11.50, df = -4.95, p = 0.0409, n = 239; Fig 2A). Staying was observed a total of 304 times (mean ± SE

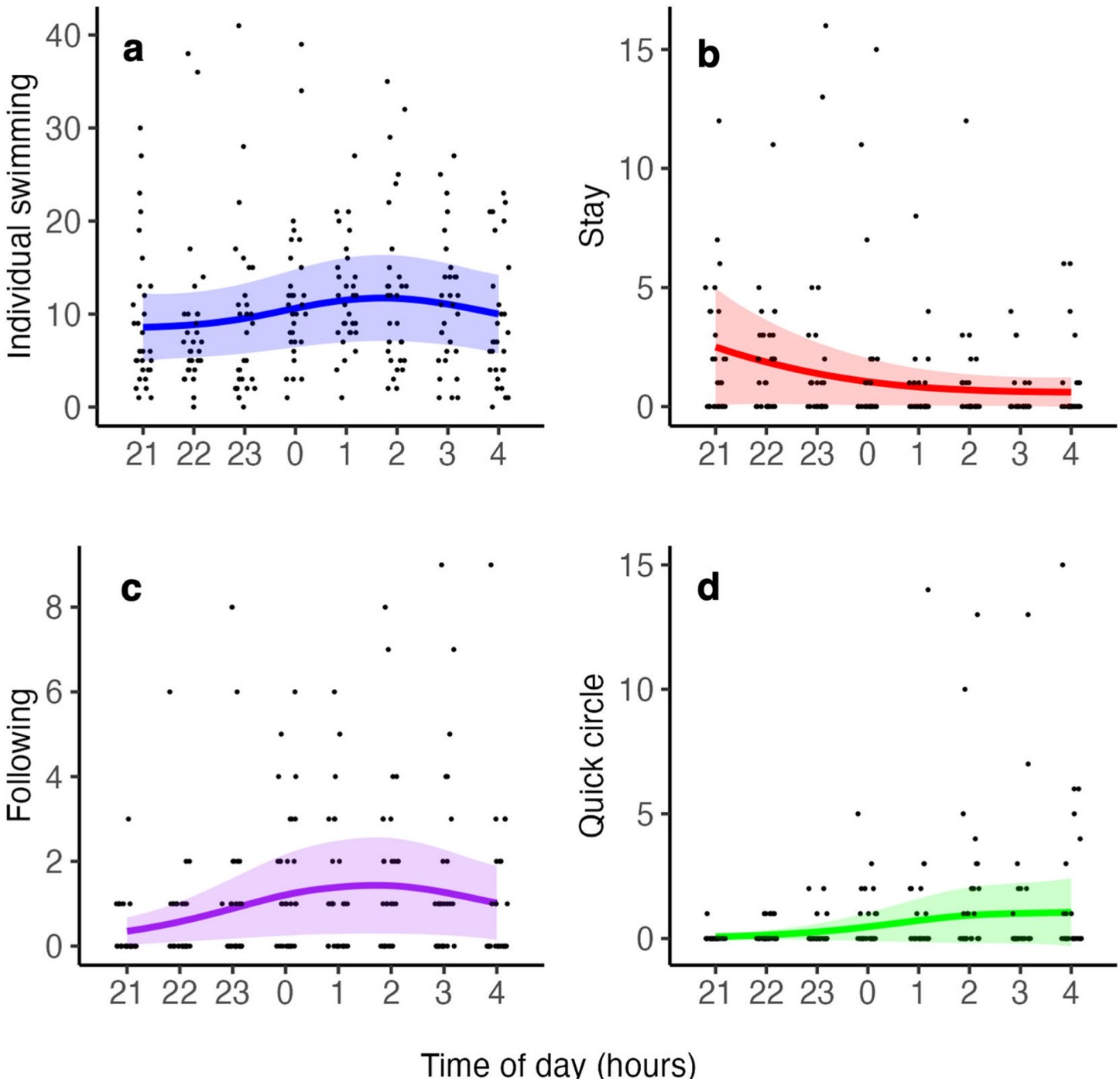

**Fig 2. Changes in nocturnal behaviors from 21:00 to 5:00 in medaka (*Oryzias latipes*) during the breeding season in Gifu, Japan.** (a) individual swimming, (b) staying, (c) following, and (d) quick circle. Each plot signifies the observed values from the analyzed videos. The regression curve was based on the generalized additive mixed model (GAMM), and the shading indicates the 95% confidence intervals. 21:00–4:00: n = 30 for each, 4:00–5:00: n = 29 video samples.

per one video = 1.27 ± 0.67, n = 239). The highest frequency was from 21:00 to 22:00, at 62 times in total (mean frequency per video ± SE = 2.07 ± 0.51, n = 30), and the lowest from 3:00 to 4:00, at 13 (mean frequency per video ± SE = 0.43 ± 0.19, n = 30). Staying

decreased from 21:00 to 1:00 and remained low from 1:00 to 5:00 (GAMM, deviance = -27.61, df = -4.02, p < 0.0001, n = 239; Fig 2B).

Following and quick circles, both courtship behaviors were observed from 21:00 to 5:00. Following was observed a total of 260 times (mean ± SE per one video = 1.09 ± 0.11, n = 239). The lowest frequency was from 21:00 to 22:00, at 9 times in total (mean frequency per video ± SE = 0.30 ± 0.12, n = 30), and the highest from 2:00 to 3:00 at 50 times (mean frequency per video ± SE = 1.67 ± 0.37, n = 30). Following gradually increased from 21:00 to 22:00 and remained high after 1:00 (GAMM, deviance = -23.12, df = -4.13, p = 0.0001, n = 239; Fig 2C). Quick circle was observed a total of 179 times (mean ± SE per one video = 0.75 ± 0.14, n = 239). The lowest frequency was from 00:00 to 22:00, at 1 time in total (mean frequency per video ± SE = 0.03 ± 0.03, n = 30), and the highest from 2:00 to 3:00, at 48 times (mean frequency per video ± SE = 1.60 ± 0.55, n = 30). Quick circle gradually increased from 23:00 and remained high after 1:00 (GAMM, deviance = -20.86, df = -3.09, p = 0.0001, n = 239; Fig 2D). No spawning behavior was observed during the video recording period.

## Discussion

Many aspects of the ecology, behavior, reproduction, and life history of model organisms, including medaka, remain unexplored in the wild. Spawning in medaka begins an hour before or after the lights are turned on [31, 35–37] Although spawning is thought to start from early to mid-morning in the wild [13], no studies have been conducted on the behavior of medaka in the wild. Our quantitative assessment of detailed behavior using video recordings revealed, for the first time, that medaka commenced spawning around midnight, several hours earlier than previously thought. Furthermore, we found that medaka activity levels increased around midnight, peaking at approximately 2:00, and courtship behavior increased around midnight and was maintained at a high level thereafter.

Previous studies have suggested that medaka spawning occurs within one hour before or after the lights are turned on [31, 35–37]. However, these studies did not directly observe spawning in the dark and instead estimated spawning times based on the developmental stages of eggs after spawning. Consequently, the actual spawning onset time could not be identified. Our nocturnal observations using videos revealed that despite sunrise occurring at approximately 5:00, females with eggs adhered to their abdomens were observed around midnight (23:54, 00:47, and 00:01) for four consecutive days. Therefore, at least at the study sites, spawning started around midnight, much earlier than previously documented.

Video analysis elucidated the transitions in nocturnal medaka behavior. From 21:00 to midnight, the activity levels were low, with many individuals sleeping or swimming slowly. The rate of slow swimming decreases as midnight approaches. In contrast, individuals engaged in courtship gradually increased from 21:00, with quick circles increasing from 23:00 and maintaining high levels until sunrise. These results suggest that medaka engage in courtship behavior from midnight onward. Previous laboratory experiments observing 24-hour activity levels in medaka reported a gradual increase in activity, particularly 4–5 h before the lights were turned on [29]. In this study, sunrise time was approximately 5:00, and our findings of increased activity of medaka in the wild at approximately 23:00 are generally consistent with previous laboratory results showing changes in activity. In addition, video observations revealed for the first time that the increase in activity before lighting was attributable to mating behavior.

Medaka may initiate mating behavior at night to avoid predation on eggs. Some species of wrasses and Japanese ornate dragonets spawn around sunset or at night, which is thought to be due to the lower activity levels of plankton predators that consume eggs and piscivorous

 

predators that target adult spawning during this period [38, 39]. This rationale may also apply to medaka. Although detailed studies on the trophic position of medaka are limited, they are mainly preyed upon by dragonfly larvae, birds, and predatory fish [40–42]. While medaka face the risk of predation by birds and piscivorous predators, the lower activity levels of these predators at night compared to daytime allow them to engage in mating behavior without needing vigilance against predators.

It has been reported that medaka can engage in mating behavior without relying on visual cues [31] and that olfaction plays a crucial role in mating [34]. Medaka also spawns in the dark [30]. Based on these insights, it is conceivable that male medaka in the wild can identify ovulated females using olfactory cues without relying on visual cues. Consistent with these findings, in the present study, we observed a temporal correlation between increased courtship around midnight and the subsequent detection of post-spawning females. It is assumed that medaka initiates courtship behavior around midnight and begins spawning with post-ovulatory females shortly after ovulation. Further research is required to determine the precise onset of ovulation in wild female medaka. Furthermore, fish use chemical cues to communicate with their surroundings and conspecifics [43–45]. Investigating the cues that male medaka use to identify spawning-ready females is expected to deepen our understanding of medaka ecology and mating behavior.

This study has two limitations. First, although many females with eggs that spawned on their abdomens were observed in our study field, no eggs were found on any of the substrates. In aquaria and artificial ponds, medaka females suspend spawned eggs on their abdomens and rub their bodies against the aquatic plants and artificial spawning beds to attach them [46]. Thus, future studies should aim to identify medaka spawning sites in the wild. By identifying spawning sites, the differences between courtship and egg attachment locations can be determined, and a deeper understanding of the reproductive behavior of medaka can be obtained. Second, we used red light for nocturnal video recordings because it is less likely to disturb the nocturnal activity of fish, including medaka [13]. It is widely used for underwater recordings at night [47]. On the contrary, however, optomotor response studies have demonstrated that several fish species, including medaka, can perceive red light [48–51]. According to a previous study on medaka, photoperiod-reversed conditions under white light cause a shift in spawning initiation time, but this effect appears 4–5 days after reversal [30]. Therefore, if red light has a large effect on medaka spawning and mating behavior as the white light, it would be expected that a similar shift in spawning time would occur in this field. In this study, however, despite the light being turned on at approximately 21:00, spawning occurred 3 h later on all four consecutive days, and no shift in the spawning time was observed. Even on the first day, when the influence of the red light was considered minimal, individuals initiated spawning at 23:54. Additionally, behavioral observations conducted in semi-natural environments using infrared illumination and cameras that are imperceptible to medaka fish indicated that medaka begin spawning and courtship at midnight (Kondo and Awata, submitted to a journal), which is consistent with the results of this study. Although no studies have quantitatively investigated the effects of red light on medaka behavior and reproduction, we speculate that red light has little effect on medaka behavior and reproduction. Further studies are required to confirm these hypotheses.

Many aspects of the ecology, behavior, reproduction, and life history of model organisms, including medaka, remain unexplored in the wild. The lack of comparative verification studies between laboratory observations and field data has led to a renewed recognition of the importance of conducting systematic field surveys and understanding their results. For example, zebrafish *Danio rerio* are widely used as model organisms in the laboratory; however, little is known about their life in the wild [52]. Consequently, for a long time, the reproductive

 

behavior and life history of zebrafish in the wild could only be inferred from laboratory observations. Recent advancements in field research have led to the accumulation of knowledge regarding the factors that influence the survival and reproduction of zebrafish in the wild [53–55]. These studies have revealed that morphology, boldness, aggression, and shoal size differ between laboratory and natural environments [56–58]. Furthermore, substantial variations in these traits have been observed in wild populations. Cichlid fish have attracted attention as models for interdisciplinary research in ecology, behavior, evolution, genomics, genetics, developmental biology, and behavioral biology [59]. The advantage of cichlids is the extensive literature available on their ecology and behavior in the wild. This has led to the accumulation of knowledge in diverse research fields, including behavior and ecosystem ecology, enabling an integrated understanding of biological organization at various levels, from genes to phenotypes, individuals, populations, and communities [60, 61].

In conclusion, this study showed that medaka in the wild initiate spawning during late nocturnal hours and exhibit vigorous courtship behavior at midnight. Future research should focus on comprehensively examining the reproductive behavior of medaka over 24 h in their natural habitat and other areas to elucidate when courtship and reproduction peak. These findings will contribute to understanding the reproduction of medaka in the laboratory and progress in different fields of study, such as physiology, genetics, and conservation. Although medaka and other model organisms are invaluable in laboratories, their ecology in the wild remains largely unknown. Further field observations are essential to compare the behavioral patterns observed in natural environments with those from laboratory experiments. This approach is expected to deepen our understanding of the mechanisms underlying the biological phenomena observed in controlled settings and their adaptive significance in nature. Furthermore, understanding the ecology and behavior of the subject species in the wild will contribute to improving animal welfare, such as reducing stress, and an appropriate research setting that considers the welfare of the species should dramatically increase the reliability of research findings.

## Supporting information

**S1 Table. Temperatures, sunrise time, and sunset time in Gifu, Japan, from July 25 to August 10, 2023.**
(PDF)

**S2 Table. Number of medaka (*Oryzias latipes*) recorded during the breeding season in Gifu, Japan.**
(PDF)

**S3 Table. Raw data for this study.**
(CSV)

**S1 Movie. Video of female medaka hanging eggs.** Females egg hanging are indicated by circular annotations. In addition to the medaka, the video captured other fish species, including bitterlings and dark chubs.
(MP4)

## Acknowledgments

We thank the members of the Laboratory of Animal Sociology, Osaka Metropolitan University, for their assistance throughout the study. We also acknowledge Editage (www.editage.jp) for the English language editing.

## Author Contributions

**Conceptualization:** Yuki Kondo, Yasunori Koya, Satoshi Awata.

**Data curation:** Yuki Kondo, Kotori Okamoto, Yuto Kitamukai.

**Formal analysis:** Yuki Kondo, Kotori Okamoto, Yuto Kitamukai, Satoshi Awata.

**Funding acquisition:** Yuki Kondo, Satoshi Awata.

**Investigation:** Yuki Kondo, Kotori Okamoto, Satoshi Awata.

**Methodology:** Yuki Kondo, Kotori Okamoto, Satoshi Awata.

**Project administration:** Yuki Kondo, Yasunori Koya, Satoshi Awata.

**Resources:** Yuki Kondo.

**Software:** Yuki Kondo.

**Supervision:** Yuki Kondo, Satoshi Awata.

**Validation:** Yuki Kondo.

**Visualization:** Yuki Kondo.

**Writing – original draft:** Yuki Kondo.

**Writing – review & editing:** Yuki Kondo, Yasunori Koya, Satoshi Awata.

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
