## [Decision Letter · Decision Letter 0]

26 Dec 2024

PONE-D-24-48646Medaka (Oryzias latipes) initiate courtship and spawning late at night: Insights from field observationsPLOS ONE

Dear Dr. Kondo,

Thank you for submitting your manuscript to PLOS ONE. After careful consideration, we feel that it has merit but does not fully meet PLOS ONE’s publication criteria as it currently stands. Therefore, we invite you to submit a revised version of the manuscript that addresses the points raised during the review process.

This manuscript has been well received by both reviewers and there are only relatively minor adjustments that are needed for the text. 

We look forward to receiving your revised manuscript.

Kind regards,

Nicholas S. Foulkes, D.Phil

Academic Editor

PLOS ONE

Journal requirements: When submitting your revision, we need you to address these additional requirements. 1. Please ensure that your manuscript meets PLOS ONE's style requirements, including those for file naming. The PLOS ONE style templates can be found at https://journals.plos.org/plosone/s/file?id=wjVg/PLOSOne_formatting_sample_main_body.pdf and https://journals.plos.org/plosone/s/file?id=ba62/PLOSOne_formatting_sample_title_authors_affiliations.pdf. 2. Thank you for stating the following in the Acknowledgments Section of your manuscript: [We thank the members of the Laboratory of Animal Sociology, Osaka Metropolitan University for their assistance throughout the study. We also acknowledge Editage (www.editage.jp) for the English language editing. This study was funded by the Japan Society for the Promotion of Science (JSPS) KAKENHI (22K20666 to Y. Kondo and 23H03868 to S. A.), Sasakawa Scientific Research Grant from the Japan Science Society (2023-5013 and 2024-5010 to Y. Kondo), and the Tokyo Zoological Park Society Wildlife Conservation Fund (to Y. Kondo)]We note that you have provided funding information that is not currently declared in your Funding Statement. However, funding information should not appear in the Acknowledgments section or other areas of your manuscript. We will only publish funding information present in the Funding Statement section of the online submission form. Please remove any funding-related text from the manuscript and let us know how you would like to update your Funding Statement. Currently, your Funding Statement reads as follows:  [This study was funded by the Japan Society for the Promotion of Science (JSPS:https://www.jsps.go.jp/) KAKENHI (22K20666 to Y. Kondo and 23H03868 to S. A.), Sasakawa Scientific Research Grant from the Japan Science Society (2023-5013 and 2024-5010 to Y. Kondo: https://www.jss.or.jp/ikusei/sasakawa/), and the Tokyo Zoological Park Society Wildlife Conservation Fund (to Y. Kondo: https://www.tokyo-zoo.net/fund/index.html). The funders had no role in the study design, data collection and analysis, decision to publish, or preparation of the manuscript] Please include your amended statements within your cover letter; we will change the online submission form on your behalf. 3. Please upload a copy of S1 Table and S2 Table to which you refer in your text on page 14. Please amend the file type to 'Supporting Information'. If the Supplementary file is no longer to be included as part of the submission please remove all reference to it within the text.

Reviewers' comments:

Reviewer's Responses to Questions

**Comments to the Author**

1. Is the manuscript technically sound, and do the data support the conclusions?

Reviewer #1: Yes

Reviewer #2: Yes

2. Has the statistical analysis been performed appropriately and rigorously? 

Reviewer #1: Yes

Reviewer #2: Yes

3. Have the authors made all data underlying the findings in their manuscript fully available?

Reviewer #1: Yes

Reviewer #2: Yes

4. Is the manuscript presented in an intelligible fashion and written in standard English?

Reviewer #1: Yes

Reviewer #2: Yes

5. Review Comments to the Author

Reviewer #1: This is a nice field study that provides novel information on the mating behaviour of a species commonly used as laboratory model. I provided below a list of suggestions that may improve the presentation of the study and in a few cases the credibility of the conclusions.

Suggestions

L27 Would not be clearer if welfare improvement was mentioned around here? Maybe also mention that the results of various types of research can be flawed by lack of welfare/presence of stress, such as for behavioural studies.

L46-47 This sentence seems to be redundant with lines 38-39. I suggest to remove it, also considering that the paragraph focus on laboratory settings.

L50 Looking for mates is a possible explanation but it is not suggested based on what written.

L65 Please specify what is this clade categorisation for readers who do not work with medaka.

L85 It is important to exclude that the red light did not trigger mating. The light could have been perceived as the beginning of sunrise by the nearby fish. From the results, I would say that mating occurred after the appearance of the light. Maybe add this conclusion in your discussion if you think it is correct.

L93 I understand that the presence of eggs was taken as evidence that spawning has just occurred. For this to be true, we need data demonstrating that the eggs are always dropped soon after spawning. Is there any supporting study to be cited in the literature?

L170 The temporal trend of the mating behaviours is similar to that of appearance. Can you compute a variable that excludes that the increase in mating behaviours was just the consequence of more fish appearing? I am thinking to something like a ratio between the number of fish observed and the number of these fish conducting mating behaviours.

Reviewer #2: The manuscript "Medaka (Oryzias latipes) initiate courtship and spawning late at

night: Insights from field observations”by Kondo et al. reports on the temporal mating behaviour of wild medaka in their native habitat.

This interesting filed work documents mating behaviour of a wild medaka group using simple and therefore reliable video observation. The results reveal a significant difference of mating time from wild medaka to those under laboratory conditions. It thus provides important information on basic biological attributes of this widely used vertebrate genetic model and underscores the necessity of field work to better know/understand our laboratory animal models.

The filed work is well documented using a simple observation methodology with a adequate analysis of the data. The manuscript is well written, but there are a few errors/weak phrasings in the abstract that should be addressed prior to publication.

minor points:

line 3: To complement laboratory experiments….rather than: to better inform…

line 6 delete “show"

line 8 change to: revealed the presence of post-spawning females….or an adequate phrasing

6. PLOS authors have the option to publish the peer review history of their article (what does this mean?). If published, this will include your full peer review and any attached files.

Reviewer #1: No

Reviewer #2: No

---

## [Author Response · Author response to Decision Letter 0]

3 Jan 2025

Reviewer #1 Comments and Responses

This is a nice field study that provides novel information on the mating behaviour of a species commonly used as laboratory model. I provided below a list of suggestions that may improve the presentation of the study and in a few cases the credibility of the conclusions.

REPLY: We sincerely thank the reviewer #1 for his/her thorough review and positive assessment of our field study on medaka mating behavior. The reviewer’s constructive suggestions have helped improve both clarity and scientific rigor of our manuscript. We have carefully addressed all points to enhance the presentation and strengthen our conclusions. Specific responses to each comment are detailed below.

Suggestions

L27 Would not be clearer if welfare improvement was mentioned around here? Maybe also mention that the results of various types of research can be flawed by lack of welfare/presence of stress, such as for behavioural studies.

REPLY: We have added a sentence to explicitly mention how stress or lack of welfare can impact research outcomes. See lines 34-37.

L46-47 This sentence seems to be redundant with lines 38-39. I suggest to remove it, also considering that the paragraph focus on laboratory settings.

REPLY: The suggested sentence has been removed to streamline the paragraph and maintain focus on laboratory settings. See line 56.

L50 Looking for mates is a possible explanation but it is not suggested based on what written.

REPLY: We have revised the text to make the statement about searching for mates more tentative and evidence-based. See lines 58-59.

L65 Please specify what is this clade categorisation for readers who do not work with medaka.

REPLY: We have added clarifying text about medaka clades. See lines 74-76.

L85 It is important to exclude that the red light did not trigger mating. The light could have been perceived as the beginning of sunrise by the nearby fish. From the results, I would say that mating occurred after the appearance of the light. Maybe add this conclusion in your discussion if you think it is correct.

REPLY: We have addressed concerns about the red light effects by:

1. Adding references to prior research showing no effects on reproductive behavior. See lines 93-95.

2. Revising the flow of discussion on light effects. See lines 243-257.

L93 I understand that the presence of eggs was taken as evidence that spawning has just occurred. For this to be true, we need data demonstrating that the eggs are always dropped soon after spawning. Is there any supporting study to be cited in the literature?

REPLY: We are sorry for insufficient information of medaka spawning. Different from many other fishes, medaka females temporarily carry the spawned eggs hanging from their abdomen. Within a few hours, they attach all these eggs to aquatic plants or similar substrates. We have added these information with supporting citations. See lines 103-109.

L170 The temporal trend of the mating behaviours is similar to that of appearance. Can you compute a variable that excludes that the increase in mating behaviours was just the consequence of more fish appearing? I am thinking to something like a ratio between the number of fish observed and the number of these fish conducting mating behaviours.

REPLY: In response to the comment, we created figure showing the proportion of each behavior observed during different time periods (see below). The results showed no clear variation in the proportion of individual swimming across time periods. However, the proportion of Stay decreased from 21:00 to 1:00, while the proportions of courtship behaviors such as Following and Quick Circle increased. This indicates that the increase in courtship behavior frequency was not merely due to the increased appearance of medaka during late-night to early-morning hours but rather reflects an actual increase in courtship behavior. As these findings are suggested in the current Fig. 1 and Fig. 2, we decided not to include the new figure in the manuscript. See Figures.

Reviewer #2 Comments and Responses

The manuscript "Medaka (Oryzias latipes) initiate courtship and spawning late at

night: Insights from field observations” by Kondo et al. reports on the temporal mating behaviour of wild medaka in their native habitat.

This interesting filed work documents mating behaviour of a wild medaka group using simple and therefore reliable video observation. The results reveal a significant difference of mating time from wild medaka to those under laboratory conditions. It thus provides important information on basic biological attributes of this widely used vertebrate genetic model and underscores the necessity of field work to better know/understand our laboratory animal models.

The filed work is well documented using a simple observation methodology with a adequate analysis of the data. The manuscript is well written, but there are a few errors/weak phrasings in the abstract that should be addressed prior to publication.

REPLY: We greatly appreciate the reviewer #2 for his/her positive evaluation of our field study. We are particularly encouraged by the recognition of how our findings contribute to understanding basic biological attributes of this important model organism and highlight the value of field observations. As suggested, we have carefully revised the abstract to address the identified phrasing issues.

minor points:

line 3: To complement laboratory experiments….rather than: to better inform…

REPLY: Revised to “To complement laboratory experiments” as suggested. See line 13.

line 6 delete “show”

REPLY: Deleted “show” for conciseness. See line 15.

line 8 change to: revealed the presence of post-spawning females….or an adequate phrasing

REPLY: Revised to “revealed the presence of post-spawning females” for clarity. See line 18.

---

## [Editor Report · Decision Letter 1]

15 Jan 2025

Medaka (Oryzias latipes) initiate courtship and spawning late at night: Insights from field observations

PONE-D-24-48646R1

Dear Dr. Kondo,

We’re pleased to inform you that your manuscript has been judged scientifically suitable for publication and will be formally accepted for publication once it meets all outstanding technical requirements.

Kind regards,

Nicholas S. Foulkes, D.Phil

Academic Editor

PLOS ONE
---

## [Editor Report · Acceptance letter]

17 Jan 2025

PONE-D-24-48646R1 

PLOS ONE

Dear Dr. Kondo, 

I'm pleased to inform you that your manuscript has been deemed suitable for publication in PLOS ONE. Congratulations! Your manuscript is now being handed over to our production team.

Kind regards, 

on behalf of

Dr. Nicholas S. Foulkes 

Academic Editor

PLOS ONE